# *i*REPO: *i*MPLICIT REWARD PAIRWISE DIFFERENCE BASED EMPIRICAL PREFERENCE OPTIMIZATION

## ABSTRACT

While astonishingly capable, large Language Models (LLM) can sometimes produce outputs that deviate from human expectations. Such deviations necessitate an alignment phase to prevent disseminating untruthful, toxic, or biased information. Traditional alignment methods based on reinforcement learning often struggle with the identified instability, whereas preference optimization methods are limited by their overfitting to pre-collected hard-label datasets. In this paper, we propose a novel LLM alignment framework named *i*REPO, which utilizes implicit Reward pairwise difference regression for Empirical Preference Optimization. Particularly, *i*REPO employs self-generated datasets judged by empirical human (or AI annotator) preference to iteratively refine the aligned policy through a novel regression-based loss function. Furthermore, we introduce an innovative algorithm backed by theoretical guarantees for achieving optimal results under ideal assumptions and providing a practical performance-gap result without such assumptions. Experimental results with Phi-2 and Mistral-7B demonstrate that *i*REPO effectively achieves self-alignment using soft-label, self-generated responses and the logit of empirical AI annotators. Furthermore, our approach surpasses preference optimization baselines in evaluations using the Language Model Evaluation Harness and Multi-turn benchmarks.

## 1 INTRODUCTION

Large Language Models (LLMs) represent a cutting-edge frontier in artificial intelligence, harnessing vast textual data to produce remarkably human-like text across diverse applications like customer service and decision support systems. Unlike traditional neural networks, training LLMs involves a multi-layered stack of processes, typically unfolds in three main phases: (1) base model training, where the foundational machine learning model, often a transformer architecture, learns from a vast dataset; (2) supervised fine-tuning, which involves refining the model with curated, high-quality datasets to enhance its performance on specific tasks; and (3) human alignment, which refines model outputs based on feedback to align more closely with human expectations and ethical standards.

Despite the sophistication of their training, LLMs can still occasionally generate outputs that are biased, untruthful, or irrelevant (Gallegos et al., 2024). This underscores the crucial need for an alignment phase to correct such deviations and ensure that the models perform ethically and effectively. Alignment strategies can be broadly categorized into *online* and *offline* approaches. Online alignment involves methods like Reinforcement Learning from Human Feedback (RLHF) integrated with Proximal Policy Optimization (PPO) (Christiano et al., 2017; Schulman et al., 2017). This dynamic approach leverages human feedback to directly influence the model's learning trajectory, capitalizing on PPO's ability to manage complex behaviours and maintain stability in updates. However, PPO's vulnerability to instability due to reward scaling and KL divergence poses further challenges in maintaining consistent performance.

Conversely, offline alignment is often referred to as preference-based learning, employing techniques such as Direct Preference Optimization (DPO) (Rafailov et al., 2023; Wang et al., 2023) and Identity Preference Optimization (IPO) (Azar et al., 2023). Such methods utilize pre-existing datasets of human preferences and integrate reward modeling directly with policy optimization, thereby eliminating the need for ongoing human interaction. However, they often oversimplify by focusing solely on the binary order between responses, potentially overlooking nuanced differences in reward

quality and leading to potential overfitting. Furthermore, the effectiveness of DPO and IPO heavily relies on the quality and diversity of the preference data. Our observations suggest that the quality difference between 'chosen' and 'rejected' responses in some offline preference dataset samples is minimal. This implies that relying solely on implicit reward differences may not be sufficient.

In this paper, we propose a novel LLM alignment framework, namely *implicit Reward Pairwise Difference Regression for Empirical Preference Optimization* ($i$REPO). By recasting the traditional human preference model as an empirical preference optimization problem, $i$REPO aims to dynamically align LLMs using their self-generated responses paired with soft labels. This method offers a distinct advantage over traditional preference optimization approaches that rely on static, pre-collected offline datasets. The core innovation of $i$REPO lies in regressing the implicit reward pairwise difference directly to the logit of human preferences, thereby eliminating the need for explicit reward model learning. $i$REPO integrates continuous policy updates and data generation with real-time querying of human feedback, potentially narrowing the gap between the model's self-generated distribution and the desired target distribution. This approach not only enhances the responsiveness of LLMs to evolving data and norms but also improves the relevance and accuracy of the outputs by ensuring they are more reflective of current human judgments and preferences.

The main contributions of this work are summarized as follows.

- We propose a novel preference optimization framework $i$REPO that regresses the implicit reward pairwise difference to the logit of empirical human (or AI annotators) preferences. The logit, derived from Zermelo rankings of multiple self-generated responses, effectively captures the quality gap between the strongest and weakest responses. This allows for the maximization of the implicit reward gap while accounting for the actual quality of responses, thereby enhancing model alignment with empirical preferences.

- We design a new loss function and propose a corresponding algorithm to solve its empirical risk minimization in $i$REPO. We provide theoretical results showing (i) an optimal alignment policy obtained by $i$REPO *under ideal conditions* and (ii) a performance gap between $i$REPO and the optimal policy *without the ideal conditions*.

- Experimental results on well-known foundation models, such as Phi-2 and Mistral-7B, affirm the superior performance of $i$REPO, showcasing its effectiveness over well-established baselines in both Language Model Evaluation Harnesses and Multi-Turn benchmarks.

## 2  RELATED WORK

**Alignment LLM through Reinforcement Learning:** Recent advances in aligning LLMs have increasingly leveraged reinforcement learning (RL) methods that incorporate human feedback directly into the learning process (Christiano et al., 2017). Ouyang et al. (2022) introduced a paradigm where RL agents learn from human feedback (RLHF) in the form of preferences between pairs of trajectory snippets instead of rewards from the environment. This method, alongside statistical gradient-following algorithms like REINFORCE (Williams, 1992; Ahmadian et al., 2024) or PPO (Schulman et al., 2017), provided robust policy gradient methods that enhance the refinement of LLMs in large action spaces and complex optimization landscapes.

**Iterative and Online Alignment**: Further building on RLHF, iterative and online methods (Xiong et al., 2024; Ye et al., 2024) developed processes that continuously align RL policies by incorporating feedback while maintaining critical characteristics of the original policies, thus ensuring both adherence to human preferences and robust policy performance. On the other hand, the application of game-theoretic concepts like minimax and Nash equilibriums has also been explored as a means to enhance the robustness of model training in the face of diverse and sometimes conflicting human feedback. Munos et al. (2023); Rosset et al. (2024) adopt a Nash learning framework, which seeks to find an equilibrium that harmonizes different objectives defined by human feedback, facilitating a balanced approach to model training. Similarly, (Swamy et al., 2024) utilize a minimax framework to minimize the maximum regret, accommodating a wide spectrum of human preferences and aiming to produce policies that perform well under the most adverse conditions. Recently, Wu et al. (2024) have explored SPPO, a self-play preference optimization framework enabling models to refine their alignment through preference-based learning objectives iteratively.

**Alignment LLM through Preference Optimization:** Beyond RL, preference optimization has emerged as a powerful approach to align LLMs with human judgments (Ziegler et al., 2020). Notable developments include DPO (Rafailov et al., 2023), showcasing the potential of directly shaping language model outputs based on human preferences and bypassing traditional reward modelling methods. Additionally, variants such as fDPO (Wang et al., 2023) expanded the methodology by incorporating diverse divergence constraints to manage a wider range of preference complexities and model uncertainties. Chowdhury et al. (2024) proposed cDPO, aiming to enhance the robustness of DPO for consistent model performance in environments characterized by noisy feedback. Zhao et al. (2023) introduced sequence likelihood calibration with human feedback (SLIC-HF), accommodating various divergence measures beyond the reverse KL divergence.

In light of the advances made by DPO and its variants, significant theoretical and practical innovations continue to contribute to the alignment of language models with human preferences (Azar et al., 2023; Ethayarajh et al., 2024b; Zhao et al., 2023; Wu et al., 2024; Liu et al., 2024; Hong et al., 2024). For instance, Azar et al. (2023) presented $\Psi$-PO, a general theoretical framework that deepens understanding of learning from human preferences. Concurrently, the KTO framework (Ethayarajh et al., 2024b) is proposed, leveraging the Kahneman-Tversky human utility function based on the psychological factors for aligning model behavior with human decision-making patterns. On the practical side, Liu et al. (2024) introduced statistical rejection sampling techniques to improve the efficiency and effectiveness of preference optimization. Hong et al. (2024) suggested ORPO, a novel approach that optimizes preferences without needing a reference model, simplifying the optimization process and broadening its applicability.

## 3 PRELIMINARIES

### 3.1 RLHF WITH EXPLICIT REWARD MODELS

The RLHF pipeline for aligning LLMs typically encompasses three main phases (Ziegler et al., 2020): (1) Supervised Fine-tuning (SFT), where a pre-trained LLM undergoes supervised learning with high-quality data tailored to specific downstream tasks; (2) Reward Modeling, a critical component for capturing human preferences effectively; and (3) RL Fine-Tuning, where the model is fine-tuned to optimize the reward model's outputs. A prevalent method within the Reward Modeling phase involves constructing an explicit reward model (Christiano et al., 2017; Ouyang et al., 2022). In this approach, a prompt $x \sim \rho$ are paired with two responses $(y_w, y_l) \sim \pi_{\text{ref}}(\cdot \mid x)$ generated under a SFT reference policy $\pi_{\text{ref}}$. A preference $(y_w \succ y_l \mid x)$ is typically annotated by humans or AI based on the Bradley-Terry (BT) model (Bradley & Terry, 1952), which derives from an underlying true reward model $r^*(y, x)$ as follows.

$$\mathbb{P}(y_w \succ y_l \mid x) = \frac{\exp(r^*(x, y_w))}{\exp(r^*(x, y_w)) + \exp(r^*(x, y_l))}. \tag{1}$$

To estimate $r^*(\cdot)$, Maximum Likelihood Estimation (MLE) techniques are often applied (Ziegler et al., 2020):

$$\widehat{r} \leftarrow \arg\max_{r \in \mathcal{R}} \mathbb{E}_{x \sim \rho, (y_w, y_l) \sim \pi_{\text{ref}}} [\log \sigma(r(x, y_w) - r(x, y_l))], \tag{2}$$

where $\sigma(\cdot)$ is a sigmoid function and $\mathcal{R}$ is a class of reward functions. Using the learned $\hat{r}$, the LLM is fined tuned with PPO (Schulman et al., 2017)

$$\max_{\theta} \mathbb{E}_{x \sim \rho, y \sim \pi_\theta(y|x)} [\hat{r}(x, y) - \beta \mathbb{D}_{\text{KL}}(\pi_\theta(y \mid x) \| \pi_{\text{ref}}(y \mid x))]. \tag{3}$$

In practice, the policy $\pi_\theta$, parameterized by $\theta \in \Theta \subset \mathbb{R}^d$, is often a transformer-based model.

Denote $\hat{\theta}$ a solution to problem (3), then the corresponding optimal policy $\pi_{\hat{\theta}}$ with respect to (w.r.t) $\hat{r}(x, y)$ will satisfy the following equation Rafailov et al. (2023)

$$\hat{r}(x, y) = \beta \log\left(\frac{\pi_{\hat{\theta}}(y|x)}{\pi_{\text{ref}}(y|x)}\right) + \beta \log Z(x), \tag{4}$$

where $Z(x) = \sum_y \pi_{\text{ref}}(y \mid x) \exp(\frac{1}{\beta}\hat{r}(x, y))$ is the partition function and $\beta$ is a scaling factor.

While PPO is a popular choice for reward modelling, it encounters instability issues as different implementations of similar reward models can produce varying outcomes. This leads to inconsistent policy performance and challenges in aligning LLMs with preferences (Rafailov et al., 2023).

### 3.2 RLHF with Implicit Reward Models

The limitations of explicit reward models, particularly those using RLHF and PPO, have paved the way for the development of implicit modeling schemes. DPO (Rafailov et al., 2023) represents a pioneering approach in this area, bypassing the reward estimate step that learns $\hat{r}(x, y)$. Instead, they observed that for any arbitrary reward estimate $\hat{r}(x, y)$ with its corresponding optimal policy $\pi_{\hat{\theta}}$:

$$\hat{r}(x, y_w) - \hat{r}(x, y_l) = \beta \left( \log \frac{\pi_{\hat{\theta}}(y_w \mid x)}{\pi_{\text{ref}}(y_w \mid x)} - \log \frac{\pi_{\hat{\theta}}(y_l \mid x)}{\pi_{\text{ref}}(y_l \mid x)} \right) \tag{5}$$

which we call the *implicit reward pairwise difference*. Substitute this difference back to (2), DPO then perform MLE to directly optimize the policy:

$$\min_{\theta} -\mathbb{E}_{(x, y_w, y_l) \sim \mathcal{D}} \left[ \log \sigma \left( \beta \log \frac{\pi_{\theta}(y_w \mid x)}{\pi_{\text{ref}}(y_w \mid x)} - \beta \log \frac{\pi_{\theta}(y_l \mid x)}{\pi_{\text{ref}}(y_l \mid x)} \right) \right] \tag{6}$$

Alternatively, methods such as IPO (Azar et al., 2023), SLiC-HF (Zhao et al., 2023), KTO (Ethayarajh et al., 2024a), and SPPO (Wu et al., 2024) learn directly from human preferences, with no explicit reward model.

## 4 *i*MPLICIT Reward pairwise based Empirical Preference Optimization (*i*REPO)

### 4.1 Empirical Human Preference Model

The primary objective in aligning LLMs with human preferences is to ensure that their outputs are ethically and socially acceptable. Consider a set of $d$ possible responses $\{y_1, y_2, \ldots, y_d\}$ produced by a SFT language model. Let $H$ represent a population of annotators, which could include humans or AI/LLM rankers. The goal is to optimize the probability model for human preference as follows:

$$\mathcal{P}^*(y_i \overset{H}{\succ} y_j \mid x) = \mathbb{P}[H \text{ prefers } y_i \text{ over } y_j \mid x] = \mathbb{E}_H[\mathbb{I}(y_i \overset{H}{\succ} y_j \mid x)] \tag{7}$$

Here, $y_i$ and $y_j$ (for $i, j = 1, \ldots, d$ and $i \neq j$) are the two competing responses among which preferences are being assessed, and $x$ is the prompt based on which $y_i$ and $y_j$ are being evaluated. $\mathcal{P}^*(y_i \overset{H}{\succ} y_j \mid x)$ represents the optimal probability that the human or annotator population prefers response $y_i$ over response $y_j$ given the context $x$. To quantify such preferences, the following BT model is often utilized:

$$\mathbb{E}_H[\mathbb{I}(y_i \overset{H}{\succ} y_j \mid x)] = \frac{e^{r^*(x, y_i)}}{e^{r^*(x, y_i)} + e^{r^*(x, y_j)}} = \sigma(r^*(x, y_i) - r^*(x, y_j)) \tag{8}$$

where $r^*(x, y_i)$, representing the strength of response $y_i$, is often referred to as the true reward model. However, accessing the true population reward model directly is not feasible. Conventional approaches often approximate this reward model through reinforcement learning techniques (Williams, 1992; Ahmadian et al., 2024; Schulman et al., 2017).

In practice, one can access a finite $h$ number of human or AI annotators to approximate the human population preference. Denote $H_k$ the $k$-th annotator sampled independently from a distribution, $k = 1, \ldots, h$. We construct an empirical human preference model as follows.

$$\mathcal{P}^h(y_i \succ y_j \mid x) = \frac{1}{h} \sum_{k=1}^{h} \mathbb{I}\{y_i \overset{H_k}{\succ} y_j\} \tag{9}$$

Assuming that the preference model (9) also follows the BT model (8). Given the impracticality of directly accessing $r^*(x, y_i)$, we introduce $r^h(x, y_i)$ as a practical approximation based on $h$ annotator preferences. Denote $w_i = e^{r^h(x, y_i)}$ ($i = 1, \cdots, d$), the pairwise preference probabilities can then be computed as:

$$\mathcal{P}^h(y_i \succ y_j \mid x) = \frac{e^{r^h(x, y_i)}}{e^{r^h(x, y_i)} + e^{r^h(x, y_j)}} = \frac{w_i}{w_i + w_j}$$

Suppose a pool of $h$ annotators (either humans or LLMs) is available to evaluate preferences between each pair of responses $(y_i, y_j)$. Let $h_{ij}$ be the number of times $y_i$ is preferred over $y_j$ among

annotators. Through analyzing the evidence from these pairwise comparisons, we can employ a maximum likelihood estimation (MLE) to optimize the weights $w_1, w_2, \ldots, w_d$ corresponding to the empirical preference ratings. The likelihood of the observed evidence given the strengths, represented by a matrix $H_e = [h_{ij}]$ and a strength vector $w = [w_i]$, respectively, is as follows.

$$\mathcal{P}(H_e|w) := \prod_{ij} \mathcal{P}^h(y_i \succ y_j \mid x) := \prod_{ij} \left( \frac{w_i}{w_i + w_j} \right)^{h_{ij}}, \tag{10}$$

The corresponding log-likelihood function is:

$$\log \mathcal{P}(H_e|w) = \sum_{ij} h_{ij} \log \frac{w_i}{w_i + w_j} = \sum_{ij} h_{ij} \log w_i - \sum_{ij} h_{ij} \log(w_i + w_j) \tag{11}$$

Differentiating with respect to $w_i$ ($\forall i$), and setting the result to zero we get:

$$\frac{1}{w_i} \sum_j h_{ij} - \sum_j \frac{h_{ij} + h_{ji}}{w_i + w_j} = 0, \tag{12}$$

which then can be rearranged to find $w_i$:

$$w_i = \frac{\sum_i h_{ij}}{\sum_j (h_{ij} + h_{ji})/(w_i + w_j)}, \tag{13}$$

**Zermelo's Model for $d$-response Ranking from Pairwise Comparison:** The formulation (13) provides a method to iteratively update the weights $w_i$ to maximize the log-likelihood, which primarily relies on Zermelo's theorem (Zermelo, 1929). Notably, it adjusts the strength of each response based on the observed preference counts $h_{ij}$. The update balances the observed wins of each response against the total contest outcomes, weighted by the sum of strengths, effectively utilizing empirical data to converge to the most likely estimates of the response strength. To find the optimal $w_i$ for all responses efficiently, we employ an accelerated variant of the traditional Zermelo algorithm proposed by Newman (2023), which offers computational efficiency and convergence guarantees (detailed in Alg. 1, lines 11–17). We also provide more details about these ranking algorithms in Appendix C.

The values of $w_i$ can then be sorted in order to give a ranking of the responses, or simply used in their raw form as a kind of rating. Based on the sorted list of $d$ responses, we then employ a pair of responses with the strongest strength and the lowest strength, denoted as $y_s$ and $y_l$, respectively. Their corresponding strengths, $w_s$ and $w_l$, are then used to estimate the implicit reward difference between these responses, according to the empirical human preference model described in equation (9).

$$\mathcal{P}^h(y_s \succ y_l \mid x) = \frac{w_s}{w_s + w_l} = \sigma(r^h(x, y_s) - r^h(x, y_l)). \tag{14}$$

Then, we have

$$r^h(x, y_s) - r^h(x, y_l)) = \log \frac{w_s}{w_s + w_l} = \text{logit}(\mathcal{P}^h(y_s \succ y_l|x)). \tag{15}$$

**Logit of Empirical Human Preference**: Using Zermelo-based rankings, $i$REPO strategically selects the strongest and weakest responses to uncover the broadest reward discrepancies, without explicit reward modeling. We quantify these using the logit of empirical preferences, a function well-suited for transforming probabilities to a comprehensive scale, capturing the nuanced differences between extreme values effectively. This focused analysis of polarities enhances our model's precision in preference assessment and ensures it closely mirrors human evaluative tendencies, accurately reflecting the perceived merits and demerits of responses.

## 4.2 $i$REPO: ALGORITHM

In this section, we propose a novel preference optimization, $i$REPO, presented in Algorithm 1. Suppose $y_s$ and $y_l$ are the strongest and weakest responses estimated by Zermelo rankings from pairwise comparisons, respectively. The gist of $i$REPO is regressing the *implicit reward pairwise difference* (5) to the *logit of empirical human preference* (15) via the loss function:

$$\ell_{i\text{REPO}}(\theta; x, y_s, y_l) := \left[ \beta \left( \log \frac{\pi_\theta(y_s|x)}{\pi_{\text{ref}}(y_s|x)} - \log \frac{\pi_\theta(y_l|x)}{\pi_{\text{ref}}(y_l|x)} \right) - \text{logit}(\mathcal{P}^h(y_s \succ y_l|x)) \right]^2 \tag{16}$$

---

**Algorithm 1** $i$mplicit Reward pairwise based Empirical Preference Optimization ($i$REPO)

---

1: **Input:**$\mathcal{D}^0 = \mathcal{D}_{\text{off}}$, $\pi_{\theta^0} = \pi_{\text{ref}}$, $h$: number of human or LLM annotators
2: **for** $t = 1, \ldots, T$ **do**
3:     Generate $\mathcal{D}^{(t)} = \{(x, y_1, \ldots, y_d) \mid x \sim \rho, (y_1, \ldots, y_d) \sim \pi_{\theta^{(t-1)}}(\cdot \mid x)\}$. Then $m$ independent training examples are randomly selected in uniform, denoted by $\hat{\mathcal{D}}^{(t)} = \{(x^{(k)}, y_1^{(k)}, \cdots, y_d^{(k)})\}_{k=1}^m \subset \mathcal{D}^{(t)}$
4:     **for** $k = 1, \ldots, |\hat{\mathcal{D}}^{(t)}|$ **do**
5:         **for** each pair of responses $(y_i^{(k)}, y_j^{(k)})$ **do**
6:             $h_{ij}^{(k)} \leftarrow$ number of human (LLM annotators) prefers $y_i^{(k)}$ over $y_j^{(k)}$
7:             $w_s^{(k)}, w_l^{(k)} \leftarrow$ ranking responses' strengths using ZERMELO_ALGORITHM with $H_e^{(k)} = [h_{ij}^{(k)}], \forall i, j$
8:         Compute $\text{logit}(\mathcal{P}^h(y_s^{(k)} \succ y_l^{(k)} | x^{(i)}))$ with $w_s^{(k)}, w_l^{(k)}$
9:     $\theta^{(t)} \leftarrow \underset{\theta}{\arg\min} \sum_{k=1}^{|\hat{\mathcal{D}}^{(t)}|} \ell_{i\text{REPO}}\left(\theta^{(t-1)}; x^{(k)}, y_s^{(k)}, y_l^{(k)}\right)$
10: **return** $\pi_{\theta^{(t)}}$ with best validation result.

---

11: **function** ZERMELO_ALGORITHM($H_e$)
12:     Initialize $C$ iterations, $w_i^{(0)} = 1 \; \forall i = 1, \ldots, d$
13:     **for** $t = 1, \ldots, C$ **do**
14:         **for** $i = 1, \ldots, d$ **do**
15:             $w_i^{(t)} \leftarrow \left(\sum_{j \neq i} h_{ij} w_j^{(t-1)} / (w_i^{(t-1)} + w_j^{(t-1)})\right) / \left(\sum_{j \neq i} h_{ji} / (w_i^{(t-1)} + w_j^{(t-1)})\right)$
16:     Sort $w_i^{(C)}$ to get the rankings of $d$ responses, then select the strongest $w_s$, and the lowest $w_l$
17:     **return** $w_s, w_l$

---

Table 1: Comparison of different preference optimization techniques.

| Methods | $\ell(x, y_s, y_l)$ | Reward Modeling | Training Data |
|---|---|---|---|
| DPO | $-\log \sigma(z_s - z_l)$ | implicit reward | offline |
| SLiC | $\max\{0, 1 - \beta(\log z_s - \log z_l)\}$ | implicit reward | offline |
| IPO | $((z_s - z_l) - 1/2)^2$ | implicit reward | offline |
| SPPO | $(z_s - 1/2)^2 + (z_l - 1/2)^2$ | relative preference reward | online batch |
| $i$REPO | $((z_s - z_l) - \text{logit}(\mathcal{P}^h(y_s \succ y_l | x)))^2$ | implicit reward | online batch |

In each iteration $t$, $i$REPO uses a set of $m$ training samples $\hat{\mathcal{D}}^{(t)} \subset \mathcal{D}^{(t)}$, which is generated by the policy $\pi_{\theta^{(t-1)}}$ (line 3), to calculate the logit of empirical preference (lines 5–8) and obtain the policy $\pi_{\theta^{(t)}}$ by minimizing the empirical loss (line 9).

We now compare $i$REPO with other popular preference optimization approaches.

**Loss function.** For the ease of comparison, let $z_s = \beta \log \frac{\pi_\theta(y_s|x)}{\pi_{\text{ref}}(y_s|x)}, z_l = \beta \log \frac{\pi_\theta(y_l|x)}{\pi_{\text{ref}}(y_l|x)}$, where in DPO, SLiC, IPO and SPPO, $y_s$ and $y_l$ are correspond to the chosen and the rejected responses, respectively. As shown in Table 1, $i$REPO is an implicit pairwise reward difference model similar to DPO and IPO, i.e., based on the term $(z_s - z_l)$. While DPO's loss is MLE, IPO uses a least square estimation (LSE) to which $\ell_{i\text{REPO}}$ is closest. The key difference is while IPO regresses all of the training samples to a constant $1/2$, $i$REPO has an updated $\text{logit}(\mathcal{P}^h(y_s \succ y_l | x))$ for each sample.

**Training data.** DPO and IPO use pre-collected offline dataset $\mathcal{D}_{\text{off}}$ for training. These approaches use hard label 1 to indicate $y_s \succ y_l$ for human alignment with $\mathcal{D}_{\text{off}}$. If $\mathcal{D}_{\text{off}}$'s coverage does not encompass the entire support of the target distribution, a disparity arises between $\mathcal{D}_{\text{off}}$ and the target distribution, potentially degrading the performance of these approaches (Liu et al., 2024). On the other hand, $i$REPO uses soft label $\mathcal{P}^h(y_s \succ y_l | x)$ to calculate the logit of empirical preference for human alignment on-the-fly. By updating new policies, generating new data, and querying the human feedback, $i$REPO can potentially close the gap between $\mathcal{D}^{(t)}$ and the target distribution, which will be analyzed in the next section.

**Empirical human or AI annotator preference**: To give feedback for LLM responses, human judgment has traditionally been considered the gold standard. However, recent advancements in LLM technology have broadened their utility, enabling them to effectively function as rankers or

annotators — particularly significant when LLMs serve as judges and provide feedback to other models. Moreover, there is a fact that LLM annotators are considerably more cost-effective than human resources and often match the reliability and accuracy of human judgments (Jiang et al., 2023; Li et al., 2023). Hence, in this work, we consider multiple LLM rankers (Jiang et al., 2023) operating collaboratively to provide feedback for responses generated by the aligned models. Such annotators allow for scalable, consistent, and rapid feedback, which is invaluable for the iterative process of model alignment. We provide more details on the AI annotators in Sec:5

### 4.3 $i$REPO: THEORETICAL RESULTS

**Assumption 4.1** (Realizability). Assume that there exists an $\theta^\star \in \Theta$ such that $\mathbb{E}_{(x,y_s,y_l)\sim\mathcal{D}}\left[\ell_{i\text{REPO}}(\theta^\star; x, y_s, y_l)\right] = 0$.

For the ease of presentation, we define $R_\theta(x, y_s, y_l) := \beta \log\left(\frac{\pi_\theta(y_s|x)}{\pi_{\text{ref}}(y_s|x)}\right) - \beta \log\left(\frac{\pi_\theta(y_l|x)}{\pi_{\text{ref}}(y_l|x)}\right)$. Define Bernoulli distributions $p^*(x, y_s, y_l) := \text{Ber}(\mathcal{P}^*(y_s \succ y_l \mid x))$ and $p_\theta(x, y_s, y_l) := \text{Ber}(\sigma(R_\theta(x, y_s, y_l)))$, which represent the human population preference and $\pi_\theta$-induced preference distributions on an arbitrary $(x, y_s, y_l)$, respectively. The following result shows that under optimal conditions, $i$REPO is aligned with human population preference. The optimal conditions require the following assumptions.

**Assumption 4.2** (Human population preference). The number of human or AI annotators $h$ is sufficiently large such that $\mathcal{P}^h \approx \mathcal{P}^*$.

**Assumption 4.3** (No data distribution disparity). There exists an $i$REPO's training iteration $\tau^\star$ such that the $\text{supp}(\mathcal{D}^{(\tau^\star)}) \approx \text{supp}(\mathcal{D}^{(\tau^\star-1)})$ due to $\pi_{\theta(\tau^\star)} \approx \pi_{\theta(\tau^\star-1)}$.

**Lemma 4.4.** *With Assumptions 4.1, 4.2, and 4.3, and denote $\theta^{(\tau^\star)}$ a solution to*

$$\min_\theta \mathbb{E}_{(x,y_s,y_l)\sim\mathcal{D}^{(\tau^\star)}}\left[\ell_{i\text{REPO}}(\theta; x, y_s, y_l)\right] \tag{17}$$

*Then $\pi_{\theta(\tau^\star)}$ is a policy that generates responses aligned with the population human preference $\mathcal{P}^*$ in expectation of a total variance distance as follows*

$$\mathbb{E}_{x\sim\rho,(y_s,y_l)\sim\pi_{\theta(\tau^\star)}(\cdot|x)}[D_{TV}\left(p^*(x, y_s, y_l)\|p_{\theta(\tau^\star)}(x, y_s, y_l)\right)] = 0. \tag{18}$$

*Furthermore, $\pi_{\theta(\tau^\star)}$ is also an optimal policy of the following problem*

$$\max_\theta \mathbb{E}_{x\sim\rho,y\sim\pi_\theta(y|x)}\left[r_{\theta(\tau^\star)}(x, y) - \beta\mathbb{D}_{\text{KL}}\left(\pi_\theta(y \mid x)\|\pi_{\text{ref}}(y \mid x)\right)\right] \tag{19}$$

*where $r_{\theta(\tau^\star)}(x, y) = \beta \log\left(\frac{\pi_{\theta(\tau^\star)}(y|x)}{\pi_{\text{ref}}(y|x)}\right) + \beta \log Z(x), \quad \forall x \sim \rho, y \sim \pi_{\theta(\tau^\star)}(\cdot|x)$.*

We provide a proof of Lemma 4.4 in Appendix A. We next show the human-alignment performance gap between a policy produced by $i$REPO and the human population preference without optimal conditions as in Lemma 4.4. For brevity, we alternatively use $\pi^*$ for the optimal policy $\pi_{\theta(\tau^\star)}$ from Lemma 4.4.

**Theorem 4.5.** *Denote $\hat{\theta}^{(t)}$ the solution to the empirical $i$REPO minimization at an iteration $t$, (line 9 of Algorithm 1) and $p_{\hat{\theta}^{(t)}}(z) := \text{Ber}(\sigma(R_{\hat{\theta}^{(t)}}(z)))$. With Assumption 4.1, we have*

$$\mathbb{E}_{z\sim\pi^*}[D_{TV}\left(p^*(z)\|p_{\hat{\theta}^{(t)}}(z)\right)] \leq O\left(\frac{1}{\sqrt{h}}\right) + O\left(\sqrt{\frac{C^{(t)}}{m}}\right), \tag{20}$$

*where $O(\cdot)$ hides some constants and $C^{(t)}$ is the concentrability coefficient defined as*

$$C^{(t)} := \sup_{\theta\in\Theta} \frac{\mathbb{E}_{z\sim\pi^*}[R_\theta(z) - \text{logit}\,\mathcal{P}^h(z)]^2}{\mathbb{E}_{z\sim\mathcal{D}^{(t)}}[R_\theta(z) - \text{logit}\,\mathcal{P}^h(z)]^2}. \tag{21}$$

We provide a proof of Theorem 4.5 in Appendix B.

*Remark* 4.6. This theorem characterizes the preference distribution gap in terms of $h$, $m$, and $C^{(t)}$, representing human, sample, and data distribution gap complexity measures, respectively. We note that the concentrability coefficient is adapted from the reinforcement learning literature Zhang (2023).

## 5 EXPERIMENTS

### 5.1 EXPERIMENTAL SETTING

**Models and Datasets:** We use Phi-2 (Hughes, 2023) and Mistral-7B (Tunstall et al., 2023b) as the pre-trained foundation models. These models undergo supervised fine-tuning on the Ultrachat-200k dataset (Ding et al., 2023) to enhance their dialogue capabilities across diverse topics. In the alignment phase, we utilize the UltraFeedback Binarized dataset (Tunstall et al., 2023b), comprising approximately 63k prompts with pairs of chosen and rejected responses. To facilitate iterative training, we uniformly sample a subset of datasets (20k for Phi-2 and 15k for Mistral-7B) from the UltraFeedback Binarized dataset in each iteration. For $i$REPO, we only use the prompts from these subsets and use the aligned models to generate 3 responses with different sampling parameters.

**Annotators:** To provide preference on the self-generated responses, we utilize a suite of LLM rankers supported by the LLM-Blender framework (Jiang et al., 2023) for comparing and annotating the outputs generated by the aligned LLM. These rankers are based on DeBERTa (He et al., 2021) trained on various high-quality and large-scale datasets with human preference annotations such as Mix-Instruct, Summarize From Feedback (Stiennon et al., 2020), Chatbot Arena Conversations (Zheng et al., 2023c). Impressively, despite their relatively small model sizes (from 0.4B - 13B), these rankers exhibit a correlation with human preferences that approach the performance of larger models (Jiang et al., 2023). These low-cost, time-efficient annotators make $i$REPO practical and robust for evaluating self-generated responses. We detail the choice of AI annotators in Appendix D.5.

**Evaluation Benchmark:** We utilize two widely recognized evaluation benchmarks: the Language Model Evaluation Harness (LM-Eval-Harness) (Gao et al., 2023) and Multi-turn Benchmark (MT-Bench) (Zheng et al., 2023b). The LM-Eval-Harness offers a transparent evaluation platform, assessing LLMs across diverse benchmarks such as ARC (Clark et al., 2018), HellaSwag (Zellers et al., 2019), MMLU (Hendrycks et al., 2021), TruthfulQA (Lin et al., 2022), Winogrande (Sakaguchi et al., 2019), and GSM8K (Cobbe et al., 2021). These tasks are designed to test different aspects of model capabilities, with models evaluated based on their accuracy and coherence in generating responses. Meanwhile, MT-Bench evaluates LLMs based on their capacity for coherent and engaging conversations, using 3.3K expert-level pairwise human evaluations of responses from commercial LLM models to 80 specific questions.

**Baselines and Implementation:** We compare our approach against three baselines: SFT, iterative DPO (Iter-DPO) (Rafailov et al., 2023; Tran et al., 2023), and iterative IPO (Iter-IPO) (Azar et al., 2023), which are widely used in the literature. Similar to our approach, DPO and IPO recast the reinforcement learning-based alignment formulation as simple loss functions to obtain implicit reward models. While these methods depend on a dataset of preferences for direct optimization, for a fair comparison, we initially utilize pre-collected response pairs and apply an empirical human preference model to train on implicit reward pairwise differences instead of relying on fixed preference outcomes. More details on datasets, benchmarks, and experimental settings are described in Appendix D.

### 5.2 MAIN RESULTS

**Performance on Language Model Evaluation Harness:** We demonstrate the capability of $i$REPO by leveraging the comprehensive suite of tasks provided by the LM-Eval-Harness for wide-ranging evaluation. As shown in Table 2, $i$REPO shows notable improvements in model alignment and performance across tasks in this benchmark. The iterative alignment approach of $i$REPO manifests in incremental gains observed from the $i$REPO-0 to $i$REPO-2 iterations for both the Phi-2 and Mistral-7B models. In its initial iteration, $i$REPO-0 utilizes responses from the preference dataset and incorporates the logit of empirical human preferences instead of relying solely on binary preferences. The performance of $i$REPO-0 is superior to those of Iter-DPO and Iter-IPO in both models. This highlights the effectiveness of $i$REPO's new loss function, which is simple, low-complexity, yet adaptive and generalizable in aligning LLMs with human-like understanding and responsiveness.

As the iterative process progresses, $i$REPO's performance is notably enhanced through training on self-generated responses supplemented with human feedback. Notably, $i$REPO consistently outperforms SFT, Iter-DPO, and Iter-IPO regarding average scores, suggesting an effective integration of training objectives that better align with the evaluation metrics. For instance, in the Phi-2 model

Table 2: Comparison of different methods using Language Model Evaluation Harness Benchmark.

| | Method | Average | ARC | HELLA SWAG | MMLU | TRUTH FULQA | WINO GRANDE | GSM8K |
|---|---|---|---|---|---|---|---|---|
| Phi-2 | SFT | 61.10 | 61.26 | 74.86 | 57.26 | 45.46 | 74.19 | 53.57 |
| | Iter-DPO-0 | 62.53 | 64.42 | 76.87 | 57.89 | 48.83 | 73.72 | 53.45 |
| | Iter-DPO-1 | 62.24 | 63.82 | 77.12 | 58.08 | 48.86 | 71.35 | 54.21 |
| | Iter-DPO-2 | 62.10 | 63.23 | 77.10 | 57.79 | 48.72 | 71.51 | 54.27 |
| | Iter-IPO-0 | 62.59 | 63.48 | 76.28 | 58.13 | 47.53 | 75.30 | 54.81 |
| | Iter-IPO-1 | 62.96 | 63.82 | 77.52 | 58.20 | 49.43 | 74.03 | 54.73 |
| | Iter-IPO-2 | 62.98 | 63.31 | 77.51 | 58.42 | 50.61 | 73.09 | 54.92 |
| | $i$REPO-0 | 63.14 | 63.23 | 76.78 | 57.56 | 51.61 | 74.74 | 54.89 |
| | $i$REPO-1 | 63.55 | 64.08 | 76.85 | 57.75 | 51.68 | 75.37 | 55.57 |
| | $i$REPO-2 | 63.72 | 64.41 | 77.20 | 57.95 | 51.68 | 75.61 | 55.48 |
| Mistral-7B | SFT | 59.01 | 60.07 | 82.36 | 61.65 | 38.88 | 76.80 | 34.27 |
| | Iter-DPO-0 | 63.51 | 63.65 | 85.35 | 63.82 | 47.14 | 79.01 | 42.08 |
| | Iter-DPO-1 | 63.53 | 63.77 | 85.37 | 63.45 | 48.23 | 79.49 | 40.86 |
| | Iter-DPO-2 | 63.54 | 63.71 | 85.37 | 63.51 | 48.29 | 79.34 | 41.03 |
| | Iter-IPO-0 | 62.67 | 63.14 | 84.37 | 63.54 | 45.35 | 79.56 | 40.03 |
| | Iter-IPO-1 | 62.99 | 63.30 | 85.15 | 63.59 | 46.07 | 80.05 | 39.78 |
| | Iter-IPO-2 | 63.09 | 63.31 | 85.42 | 63.48 | 46.19 | 80.20 | 39.96 |
| | $i$REPO-0 | 64.25 | 65.19 | 85.37 | 62.50 | 51.85 | 79.87 | 40.71 |
| | $i$REPO-1 | 64.78 | 66.04 | 85.69 | 62.68 | 53.46 | 80.11 | 40.69 |
| | $i$REPO-2 | 65.27 | 66.64 | 85.40 | 62.68 | 55.47 | 80.74 | 40.69 |

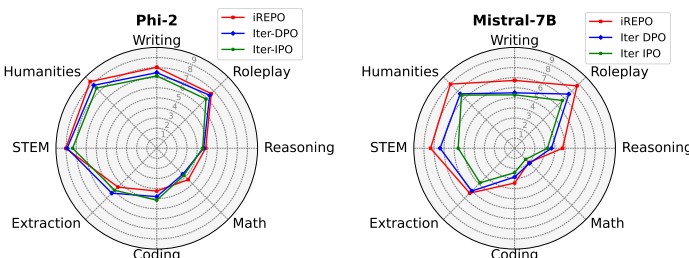

Figure 1: MT-Bench single-grading evaluation for Phi-2 and Mistral-7B models with different methods.

evaluations, $i$REPO-1 achieves a higher average score than earlier iterations and baseline methods, with significant improvements noted in the ARC, HellaSwag and TruthfulQA benchmarks. This suggests that $i$REPO's methodology enhances general performance and specifically improves the model's ability to handle complex reasoning and truthfulness in responses—a critical aspect in practical applications of LLMs. Similarly, for the Mistral-7B model, $i$REPO-1 to $i$REPO-2 shows marked improvements over the baselines in the ARC, TruthfulQA, and Winogrande tasks, reinforcing the method's utility in enhancing the understanding of context and factual accuracy. DPO and IPO are not designed for iterative training, thus slightly improving performance over iterations.

**Single-Grading on Multi-turn Benchmark:** We compare the performance of $i$REPO with other baselines in eight domains using the MT-Bench single-grading scheme. As shown in Fig. 1 and Table 3, in both Phi-2 and Mistral-7B, $i$REPO demonstrates superior performance across various domains, surpassing the best version of Iter-DPO and Iter-IPO. Particularly, in a 2-turn evaluation, $i$REPO-2 consistently achieves higher grades than others. In fields requiring an understanding of language and context, such as Writing and Humanities, our method excels by producing responses that are not only contextually appropriate but also rich in detail and coherence. This showcases $i$REPO 's capabilities in complex narrative generation applications and interactive educational content. Moreover, in technical domains like STEM and Extraction, $i$REPO proves effective, highlighting its ability to handle tasks requiring precision and high informational accuracy.

**Pairwise Winrate on Multi-turn Benchmark:** We employ the pairwise win-rate comparison method within MT-Bench, conducting more than 300 matches across 80 questions with GPT-4 serving as the judge to assess the quality of responses from competing models. In these competitions, $i$REPO-2 stands out by achieving the highest win rates against models aligned with Iter-DPO and Iter-IPO. Particularly, In Mistral-7B, $i$REPO-2 significantly outperforms other baselines with an adjusted win rate of 63.06%, compared to 54.97% for Iter-DPO and 32.05% for Iter-IPO. A similar result is observed in Phi-2, where $i$REPO-2 consistently outperforms the baselines. This underscores its ability to generate responses that are accurate, contextually relevant, and of better quality.

Table 3: Comparison of different methods on MT-Bench (win rates are adjusted following Zheng et al. (2023a)).

| Model | Method | Pairwise Comparison | | | | Single Grading Score | | |
|---|---|---|---|---|---|---|---|---|
| | | Win | Tie | Loss | Win Rate (%) | 1st Turn | 2nd Turn | Average |
| Phi-2 | Iter-DPO | 55 | 45 | 218 | 51.57 | 7.19 | 5.82 | 6.51 |
| | Iter-IPO | 40 | 66 | 214 | 45.94 | 7.01 | 5.56 | 6.28 |
| | *i*REPO-2 | 63 | 47 | 208 | 52.52 | 7.15 | 5.95 | 6.55 |
| Mistral-7b | Iter-DPO | 104 | 73 | 135 | 54.97 | 5.07 | 5.46 | 5.03 |
| | Iter-IPO | 39 | 151 | 122 | 32.05 | 4.58 | 5.02 | 4.8 |
| | *i*REPO-2 | 128 | 47 | 135 | 63.07 | 6.77 | 5.57 | 6.17 |

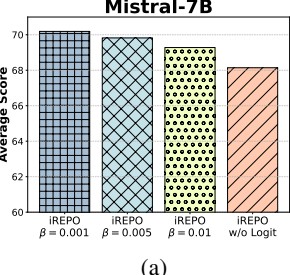 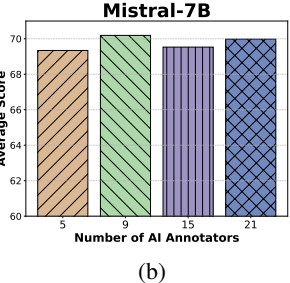

(a)           (b)

Figure 2: (a) Performance of *i*REPO with and without the logit of empirical human preference, and (b) Performance of *i*REPO with different number of AI annotators

### 5.3 ABLATION STUDIES

**Effects of $\beta$ and Pairwise Reward Difference Models:** We investigate the impact of the logit of empirical human preference in *i*REPO's loss function on the alignment performance, contrasting training with and without this feature (*i*REPO w/o Logit). The results depicted in Fig. 2a reveal that omitting the logit term significantly degrades performance. Specifically, *i*REPO w/o Logit exhibits a substantial decline in average performance. This suggests that the logit of the empirical human preference model plays a vital role in *i*REPO's loss function, steering the training process toward more effective outcomes. Additionally, our evaluation of different $\beta$ values indicates that performance variations are minimal. The model with $\beta = 0.001$ achieves the best performance, while other values of $\beta$ slightly underperform but still maintain satisfactory performance.

**Effects of Annotators:** To investigate the impact of AI annotator quantity on the performance of *i*REPO, we trained the last iteration model using different groups of annotators, with each group consisting of $\{5, 9, 15, 21\}$ members. Each group evaluates a consistent set of responses generated by the aligned model, and their feedback is incorporated to refine the model's alignment. The results in Fig. 2b demonstrate that increasing the number of annotators from 5 to 21 has minimal impact on *i*REPO's performance. Notably, models trained with 9 annotators slightly outperform those trained with different numbers of annotators. This indicates that a moderate number of AI annotators is sufficient to capture a broad range of perspectives, effectively simulating a comprehensive empirical human preference model. Furthermore, LLM annotators with proven judgment capabilities provide high-quality feedback, offering scalable, consistent, and efficient alternatives to human resources.

### 6 CONCLUSION

In this paper, we introduced *i*REPO, a novel LLM alignment framework, to address the challenges of traditional alignment methods such as instability in reinforcement learning approaches and overfitting in preference optimization methods. By utilizing implicit reward pairwise difference model and empirical preference data from self-generated responses labeled by humans or AI annotators, *i*REPO iteratively refines LLM policies through a novel regression-based loss function. This innovative approach is supported by theoretical guarantees that ensure optimal results under specific, albeit unreal, assumptions and offers practical insights into reducing performance gaps in more typical scenarios. Experimentally, we show that *i*REPO effectively implements self-alignment with Phi-2 and Mistral-7B, delivering superior performance compared to traditional preference optimization baselines in assessments using the LLM Evaluation Harness and Multi-turn benchmarks.

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

## A    PROOF OF LEMMA 4.4

*Proof.* We alternatively use $z$ and $(x, y_s, y_l)$ for brevity.

Define $\mathcal{P}^*(z) := \mathcal{P}^*(y_s \succ y_l \mid x)$, we have $\mathcal{P}^*(z) = \sigma(\text{logit } \mathcal{P}^*(z))$. Therefore, the Bernoulli distribution $p^*(z) = \text{Ber}(\mathcal{P}^*(z)) = \text{Ber}(\sigma(\text{logit } \mathcal{P}^*(z)))$.

$$\mathbb{E}_{z \sim \mathcal{D}^{(\tau^\star)}}[D_{TV}(p_{\theta^{(\tau^\star)}}(z) \| p^*(z))] \leq 2\,\mathbb{E}_{z \sim \mathcal{D}^{(\tau^\star)}}|\sigma(R_{\theta^{(\tau^\star)}}(z)) - \sigma(\text{logit } \mathcal{P}^*(z))| \quad (22)$$

$$\leq \frac{1}{2}\mathbb{E}_{z \sim \mathcal{D}^{(\tau^\star)}}|R_{\theta^{(\tau^\star)}}(z) - \text{logit } \mathcal{P}^*(z)|$$

$$\leq \frac{1}{2}\left(\mathbb{E}_{z \sim \mathcal{D}^{(\tau^\star)}}\left[R_{\theta^{(\tau^\star)}}(z) - \text{logit } \mathcal{P}^*(z)\right]^2\right)^{\frac{1}{2}}$$

$$= 0$$

where the first inequality is because by the total variance distance between any two Bernoulli distributions $p_1$ and $p_2$ defined by $\text{Ber}(\sigma(x_1))$ and $\text{Ber}(\sigma(x_2))$, $\forall x_1, x_2 \in \mathbb{R}$, respectively:

$$D_{TV}(p_1 \| p_2) = |\sigma(x_1) - \sigma(x_2)| + |1 - \sigma(x_1) - (1 - \sigma(x_2))|$$
$$= 2|\sigma(x_1) - \sigma(x_2)|.$$

The second inequality is by mean value theorem, that there exists a $x_0 \in [x_1, x_2]$ such that

$$|\sigma(x_1) - \sigma(x_2)| = \left.\frac{d\sigma}{dx}\right|_{x_0}|x_1 - x_2|$$

$$= \sigma(x_0)(1 - \sigma(x_0)|x_1 - x_2|$$

$$\leq \frac{1}{4}|x_1 - x_2|.$$

The third inequality is due to Cauchy-Schwarz. The last inequality is because $\theta^{(\tau^\star)}$ is a solution to $\min_\theta \mathbb{E}_{(x, y_s, y_l) \sim \mathcal{D}^{(\tau^\star)}}[\ell_{i\text{REPO}}(\theta; x, y_s, y_l)]$ and the Assumption 4.1. The result (18) follows because $\mathcal{D}^{(\tau^\star)} = \{(x, y_s, y_l) \mid x \sim \rho, (y_s, y_l) \sim \pi_{\theta^{(\tau^\star-1)}}\}$ and we assumed $\pi_{\theta^{(\tau^\star-1)}} \approx \pi_{\theta^{(\tau^\star)}}$.

We next show the result (19). Define a reward function

$$r_{\theta^{(\tau^\star)}}(x, y) = \beta \log\left(\frac{\pi_{\theta^{(\tau^\star)}}(y|x)}{\pi_{\text{ref}}(y|x)}\right) + \beta \log Z(x), \quad \forall x \sim \rho, y \sim \pi_{\theta^{(\tau^\star)}}(\cdot|x) \quad (23)$$

According to (3) and (4), $\theta^{(\tau^\star)}$ is also the solution to the following problem

$$\max_\theta \mathbb{E}_{x \sim \rho, y \sim \pi_\theta(y|x)}[r_{\theta^{(\tau^\star)}}(x, y) - \beta \mathbb{D}_{KL}(\pi_\theta(y \mid x) \| \pi_{\text{ref}}(y \mid x))] \quad (24)$$

and thus $\pi_{\theta^{(\tau^\star)}}$ is an optimal policy. $\qquad\square$

## B    PROOF OF THEOREM THEOREM 4.5

*Proof.* Denote $p_h(z) := \text{Ber}(\sigma(\text{logit } \mathcal{P}^h(z)))$. By triangle inequality:

$$D_{TV}(p^*(z) \| p_{\hat{\theta}^{(t)}}(z)) \leq D_{TV}(p^*(z) \| p_h(z)) + D_{TV}(p_h(z) \| p_{\hat{\theta}^{(t)}}(z)), \forall z. \quad (25)$$

We bound the right-hand side (RHS) terms on the above inequality. First,

$$D_{TV}(p^*(z) \| p_h(z)) = 2\left|\frac{1}{h}\sum_{i=1}^h \mathbb{I}(y_s \overset{H_i}{\succ} y_l) - \mathbb{E}_H[\mathbb{I}(y_s \overset{H}{\succ} y_l)]\right|$$

$$\leq O\left(\frac{1}{\sqrt{h}}\right) \quad \forall z.$$

due to the uniform law of large number. Second,

$$
\begin{aligned}
\mathbb{E}_{z \sim \pi^*}[D_{TV}\left(p_{\hat{\theta}^{(t)}}(z) \| p_h(z)\right)] &\le 2 \mathbb{E}_{z \sim \pi^*}\left|\sigma(R_{\hat{\theta}^{(t)}}(z)) - \sigma(\operatorname{logit} \mathcal{P}^h(z))\right| \\
&\le \frac{1}{2}\mathbb{E}_{z \sim \pi^*}\left|R_{\hat{\theta}^{(t)}}(z) - \operatorname{logit} \mathcal{P}^h(z)\right| \\
&\le \frac{1}{2}\left(\mathbb{E}_{z \sim \pi^*}\left[R_{\hat{\theta}^{(t)}}(z) - \operatorname{logit} \mathcal{P}^h(z)\right]^2\right)^{\frac{1}{2}} \\
&\le \frac{1}{2}\left(C^{(t)} \mathbb{E}_{z \sim \mathcal{D}^{(t)}}\left[R_{\hat{\theta}^{(t)}}(z) - \operatorname{logit} \mathcal{P}^h(z)\right]^2\right)^{\frac{1}{2}} \\
&\le O\left(\sqrt{\frac{C^{(t)}}{m}}\right),
\end{aligned}
\tag{26}
$$

where the first three inequalities are similar to those of (22). The fourth inequality is by definition (21). The last inequality is due to the concentration result of least square regression with realizability Assumption 4.1 (Zhang, 2023, Examples 3.18 and 3.25).

$\square$

## C  ZERMELO-BASED RANKINGS FROM PAIRWISE COMPARISONS

### C.1  RELATION BETWEEN ZERMELO RANKINGS AND BRADLEY-TERRY MODELS

Zermelo Rankings are based on the concept of comparing pairs of items to determine a ranking (Zermelo, 1929), which is particularly suitable for scenarios involving subjective evaluations or competitions, such as ranking outputs from a large language model. In this context, Zermelo Rankings use the Bradley-Terry model, which is a probabilistic model used to estimate the relative strength ($w_i$) of each item. For two items $i$ and $j$, the probability that item $i$ is preferred over item $j$ is given by:

$$
\mathcal{P}(i \succ j) = \frac{w_i}{w_i + w_j}
\tag{27}
$$

This model is powerful because it translates qualitative pairwise preferences into a quantitative measure of strength or skill for each item. The strengths are iteratively updated to find the maximum likelihood estimates that best explain the observed data. By utilizing pairwise comparison data, Zermelo Rankings provide an efficient mechanism to rank items even when the number of comparisons is sparse or inconsistent, which often occurs with language model evaluations.

### C.2  TRADITIONAL ZERMELO'S ALGORITHM

The traditional Zermelo algorithm (Zermelo, 1929), iteratively updates the strengths $w_i$ of each item to maximize the likelihood of observed pairwise outcomes. Given $d$ items and $h_{ij}$ is the number of times item $i$ beats item $j$, the update rule for each strength is:

$$
w_i' = \frac{\sum_{j=1}^d h_{ij}}{\sum_{j=1}^d (h_{ij} + h_{ji})(w_i + w_j)})
\tag{28}
$$

This iterative formula seeks to adjust $w_i, \forall i$ such that it accurately reflects the empirical likelihood of item $i$ beating item $j$ across multiple pairwise comparisons. The log-likelihood function that the algorithm aims to maximize is:

$$
\log P(H|w) = \sum_{ij} h_{ij} \log \frac{w_i}{w_i + w_j} = \sum_{ij} h_{ij} \log w_i - \sum_{ij} h_{ij} \log(w_i + w_j)
\tag{29}
$$

Differentiating this log-likelihood with respect to $\pi_i$ and setting the derivative to zero leads to the iterative update formula, which aims to maximize the likelihood of the given pairwise data. However, the problem with the traditional algorithm is its slow convergence, especially when applied to large-scale datasets. The convergence rate is highly sensitive to the initial values chosen for $w_i$, and the iteration can take many steps to reach stability, making it computationally inefficient.

### C.3 ACCELERATED ZERMELO'S ALGORITHM

To address the inefficiencies of the traditional approach, Newman (2023) proposed an enhanced iterative algorithm. This new method modifies the update formula to better exploit the pairwise comparison data and thereby accelerate convergence. The enhanced update rule is given by:

$$w_i' = \frac{\sum_{j=1}^{d} h_{ij} w_j (w_i + w_j)}{\sum_{j=1}^{d} h_{ji} (w_i + w_j)} \tag{30}$$

The key difference here is the introduction of weighted terms that more directly adjust the strengths based on the relative competitive outcomes between items. The numerator $\sum_{j=1}^{d} \frac{w_{ij} \pi_j}{\pi_i + \pi_j}$ takes into account both the number of wins and the relative strength of item $j$ compared to $i$, while the denominator adjusts for losses in a similar manner.

From a mathematical perspective, this update method provides a more dynamic weighting of the strength parameters, leading to several important improvements:

- *Faster Convergence*: By directly incorporating the current strengths ($w_i$ and $w_j$) into the weight adjustment, the algorithm accelerates the convergence process. This ensures that items with higher empirical strength are updated more aggressively, allowing the estimates to reach a stable state much faster than the traditional Zermelo method.

- *Reduction in Iterations*: Empirical studies have shown that the enhanced algorithm can be over a hundred times faster in some scenarios (Newman, 2023). The reduced number of iterations is particularly beneficial for ranking LLM responses, making computational efficiency crucial.

The enhanced iterative formula preserves the concavity of the likelihood function, which guarantees convergence to the global maximum, provided the network of comparisons is strongly connected (i.e., there exists a path through the network that connects every item). To prove convergence, consider the asynchronous version of the update, where a single $w_i$ is updated at each step, while others remain fixed. The log-likelihood is strictly increasing with each update unless a fixed point is reached, thereby ensuring convergence. See Newman (2023, Section 3) for more details.

### C.4 TOY EXAMPLE

We consider a toy example with three responses evaluated by a pool of 9 annotators (either human or language model outputs) who compare each pair of responses and indicate their preference. The preference matrix obtained from this evaluation is depicted in Fig. 3a. In this matrix, each element $h_{ij}$ represents the number of times response $y_i$ was preferred over response $y_j$ among all annotators. For instance, response 1 ($y_1$) was preferred 6 times over both response 2 ($y_2$) and response 3 ($y_3$), whereas response 2 was preferred 5 times over response 3. The diagonal elements are zero since a response cannot be compared with itself.

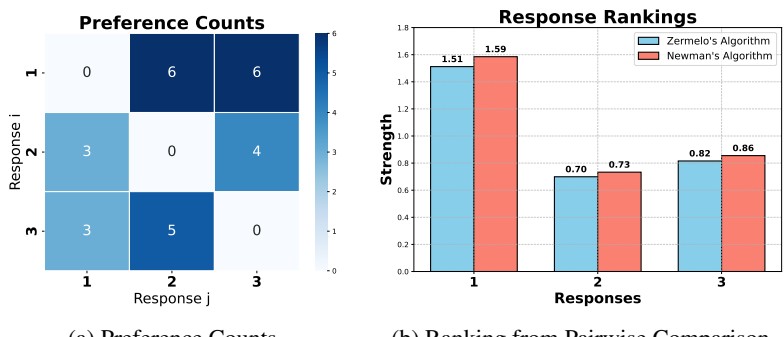

(a) Preference Counts        (b) Ranking from Pairwise Comparison

We applied both the traditional Zermelo Algorithm and Newman's Enhanced Iterative Algorithm to determine the strengths of these three responses based on the pairwise preference data. The traditional Zermelo Algorithm required 18 iterations to converge, while the enhanced algorithm converged in only 8 iterations. Both methods provide similar strength values and reasonable rankings, as shown in Fig. 3b. Notably, response 1 has the highest strength, reflecting its strong preference by annotators,

while response 2 has the lowest strength, indicating it was less preferred compared to the other responses. The example illustrates the practical benefits of the Zermelo-based approach, making it a suitable choice for evaluating and ranking responses in AI alignment and large-scale pairwise comparison settings.

# D  EXPERIMENT DETAILS

## D.1  DATASETS

**UltraFeedback-binarized (Tunstall et al., 2023b)**: is a pre-processed version of the UltraFeedback dataset, which is a large-scale, fine-grained, diverse preference dataset, used for developing and refining LLMs models focused on preference-based learning. This dataset contains approximately 64k prompts derived from a diverse array of sources, including UltraChat, ShareGPT, Evol-Instruct, TruthfulQA, FalseQA, and FLAN. Each record includes a pair of model-generated responses: one "chosen" and one "rejected," accompanied by their respective scores. These responses are selected based on preference scoring that assesses criteria like relevance, accuracy, and utility, with the "chosen" response typically having the highest overall score to reflect superior quality or better alignment with human judgment, while the "rejected" response illustrates less preferred options.

For the initial training of our algorithm, referred to as $i$REPO-0, we utilize the responses from this dataset. However, instead of employing a deterministic binary (win-lose) relationship as seen in DPO, IPO or KTO, we compute the logit of empirical human preferences using the scores of the chosen and rejected responses. These logits are then integrated into our loss function to ensure a more nuanced model training process and to facilitate fair comparisons with baseline models.

In the next training iterations ($i$REPO-1 and $i$REPO-2), we randomly pick 15k prompts and 20k prompts for Mistral-7B and Phi-2, respectively. Then, we use our aligned model to generate repsonse pairs and employ humans or AI annotators to give feedback.

## D.2  BENCHMARKS

**Language Model Evaluation Harness (LM-Eval-Harness) (Gao et al., 2023)**: serves as a structured and transparent platform for assessing the capabilities of language models across a diverse range of benchmarks. Each benchmark within this harness targets specific aspects of a language model's abilities, from reasoning and understanding to knowledge application and truthfulness in responses. In this study, we utilize the following datasets from LM-Eval-Harness for evaluation:

1. *ARC (AI2 Reasoning Challenge - 25-shot)* (Clark et al., 2018): This dataset comprises 7,787 authentic, grade-school level, multiple-choice science questions that are intended for question-answering. It is mainly used to assess the model's capacity to engage in complex reasoning.

2. *HellaSwag (10-shot)* (Zellers et al., 2019): This dataset is created to test the model's ability to predict logical scenario completions, demanding a strong sense of commonsense reasoning and contextual awareness.

3. *MMLU (Massive Multitask Language Understanding - 5-shot)* (Hendrycks et al., 2021): This dataset is designed to evaluate the model's understanding and application of knowledge across a wide range of 57 tasks, including topics such as elementary mathematics, US history, computer science, law, and more.

4. *TruthfulQA (0-shot)* (Lin et al., 2022): This dataset is specifically constructed to test the model's capability to produce responses that are accurate, truthful, and non-misleading, with a focus on ethical considerations in AI outputs. It includes 817 questions across 38 categories, such as health, law, finance, and politics.

5. *Winogrande (5-shot)* (Sakaguchi et al., 2019): This dataset contains approximately 44k problems, formulated as a fill-in-a-blank task with binary options. Its goal is to choose the right option for a given sentence which requires commonsense reasoning.

6. *GSM8k (Grade School Math 8k)* (Cobbe et al., 2021): This component assesses the model's problem-solving skills in basic arithmetic and algebra, reflecting its numerical reasoning capabilities.

Models are evaluated based on their accuracy and coherence in generating responses across these tasks, with an aggregated "Average" score providing a holistic view of their overall proficiency.

**Multi-turn Benchmark (MT-Bench) (Zheng et al., 2023b):** is crafted to evaluate how effectively language models handle multi-turn dialogues, focusing on the nuanced aspects of conversational AI, such as the ability to maintain context over several turns, the coherence and relevance of responses, and the adaptive capacity of models to shift strategies based on dialogue progression. MT-Bench comprises:

1. *Expert-Level Pairwise Human Preferences*: This component involves 3,300 pairwise comparisons conducted by experts, assessing model responses to 80 unique questions. These questions are designed to be representative of real-world conversational challenges.

2. *Participating Models*: The benchmark tests several advanced models including GPT-4, GPT-3.5, Claud-v1, Vicuna-13B, Alpaca-13B, and LLaMA-13B, providing a comparative analysis of their performance.

3. *Annotator Expertise*: The responses are evaluated by graduate students specializing in the relevant question topics, ensuring that the assessments are both knowledgeable and contextually informed.

### D.3 HYPERPARAMETERS AND IMPLEMENTATION DETAILS

**Implementation Detail:** Our implementation of $i$REPO leverages well-established frameworks and libraries to ensure robust alignment and performance enhancements across language models. The frameworks utilized include the Alignment Handbook (Tunstall et al., 2023a), TRL (von Werra et al., 2020), vLLM (Kwon et al., 2023) and Alpaca-Farm (Dubois et al., 2023) frameworks, each contributing uniquely to facilitate end-to-end training process as follows.

- **Alignment Handbook** (Tunstall et al., 2023a): e utilize this comprehensive codebase to align LLMs with human and AI preferences. It provides essential recipes and configurations, which are foundational for training $i$REPO and other baselines.

- **Transformer Reinforcement Learning (TRL)** (von Werra et al., 2020): This powerful toolset facilitates the fine-tuning and alignment of LLMs, supporting various methods like Direct Preference Optimization (DPO) and Identity Preference Optimization (IPO). We integrate $i$REPO within this framework to enhance consistency and reproducibility in our results.

- **vLLM**(Kwon et al., 2023): A library designed to accelerate LLM inference. We leverage this library to generate responses after each training iteration. vLLM achieves approximately 24 times higher throughput compared to the conventional generation method of HuggingFace Transformers (HF).

- **LLM-Blender** (Jiang et al., 2023): An ensembling framework that is designed to fuse the strength of multiple open-source LLMs to produce a confidential rank for the multiple output candidates through a pairwise comparison method which uses cross-attention to encode the input text and a pair of candidates.

**Training Details:** We enhance training efficiency by incorporating LoRA (Hu et al., 2022) with acceleration technologies such as DeepSpeed-Zero3 (Rasley et al., 2020) and FlashAttention (Dao et al., 2022). Our experiments are primarily conducted on two workstations including: an Intel® Xeon® W-3335 Processor, 512GB RAM, and 4 NVIDIA GeForce RTX 4090 GPUs; and an AMD Ryzen 3970X Processor with 64 cores and 256GB of RAM and 4 NVIDIA GeForce RTX 3090 GPUs.

**Hyperparameter Tuning**: For $i$REPO, we conducted three training iterations in total. In each iteration, we selected the model that performed best after the first epoch of training on 20k prompts from UltraFeedback to proceed to the next iteration. Due to resource limitations, we utilize LoRA Hu et al. (2022) to fine-tune our models.

Table 4: Model Traning Parameters

| Part | Hyperparameters | Mistral-7B | Phi-2 |
|---|---|---|---|
| LoRA | Rank | 128 | 256 |
| | $\alpha$ | 128 | 256 |
| | Dropout | 0.05 | 0.05 |
| Training Arguments | $\beta$ | 0.001 | 0.001 |
| | Optimizer | AdamW | AdamW |
| | Batch size/GPU | 2 | 4 |
| | Learning rate | 5.0e-6 | 5.0e-6 |
| | Training epochs | 1 | 1 |
| | Warmup ratio | 0.1 | 0.1 |
| | Schedule | cosine | cosine |
| | Gradient accumulation | 2 | 4 |
| | Max prompt length | 1024 | 1024 |
| | Max response length | 1024 | 1024 |
| | Data type | bfloat16 | bfloat16 |

### D.4 RESPONSE GENERATION

Our experiments leverage the vLLM's efficient memory sharing capabilities during response genera-
tion. Specifically, vLLM supports parallel sampling, where multiple output sequences are generated
from a single prompt. This approach allows the computational resources and memory allocated for
the prompt to be shared across different output sequences, enhancing efficiency.

For each prompt, we utilized vLLM to generate two distinct responses. The quality and variability of
these responses are influenced by sampling parameters such as max tokens, temperature, and top_p,
which are detailed as follows:

Table 5: Sampling parameters for generating responses using vLLM

| Parameter | Response 1 | Response 2 | Response 3 | Description |
|---|---|---|---|---|
| Max Tokens | 512 | 512 | 512 | The maximum length of the generated response in tokens. |
| Temperature | 0.8 | 1.0 | 0.8 | The randomness in prediction; lower values lead to more predictable text, higher values produce more varied outputs. |
| Top_p | 1.0 | 1.0 | 0.8 | The threshold for cumulative probability for selecting possible next words, allowing for a diverse set of responses. |

### D.5 CHOICE OF ANNOTATORS

In our paper, we leverage a number of LLM rankers to give preference feedback to responses
by $i$REPO across iteration. Unlike human preference annotation, which can be expensive and
time-consuming, using LLM rankers significantly reduces costs. To construct a diverse pool of
annotators, we utilize multiple open-source LLM ranker models, such as *llm-blender/pair-ranker*, *llm-
blender/PairRM* (Jiang et al., 2023) and its variants fine-tuned on different human feedback datasets,
*OpenAssistant/reward-model-deberta-v3-large-v2* (Assistant, 2023), *openbmb/UltraRM-13b* (Cui
et al., 2023), *berkeley-nest/Starling-RM-7B-alpha* (Zhu et al., 2023), etc. Despite being relatively
smaller in size, these models demonstrate strong correlations with human preferences and approach
the performance level of GPT-4, making them effective and efficient alternatives for high-quality
ranking tasks (Jiang et al., 2023).

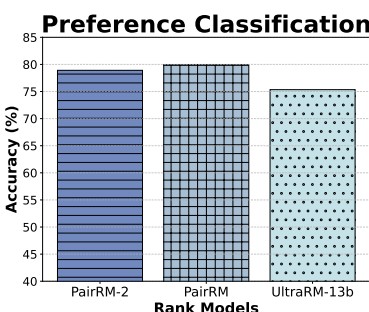

Figure 4: Preference classification accuracy of LLM rankers on Ultrafeedback-Binarized dataset.

To evaluate the performance of these LLM rankers, we utilize two models with distinct configurations: *llm-blender/PairRM* and *openbmb/UltraRM-13b*, to assess preferences between response pairs in the Ultrafeedback-Binarized dataset Tunstall et al. (2023b), which contains responses pre-labeled as 'chosen' and 'rejected'. As depicted in Fig. 4, the accuracy of the LLM rankers in preferring 'chosen' responses over 'rejected' ones is relatively high (approximately 80%), indicating their adequacy for $i$REPO and offering a cost-effective alternative to high-cost human annotators or commercial LLMs. Furthermore, we observed that there are not much gap in the quality between 'chosen' and 'rejected' responses in some samples of this dataset. This highlights that relying solely on implicit reward differences, as in methods like DPO and IPO, may not be sufficient. In contrast, leveraging logits from empirical preferences with $i$REPO provides a more robust and generalizable approach.

Another cost-effective option is to employ the pairwise annotators API wrapper provided in Alpaca-Farm (Dubois et al., 2023) and AlpacaEval (Dubois et al., 2024). This API facilitates pairwise feedback from commercial LLM models (e.g. GPT-4, ChatGPT).

In most of our experiments, we utilize nine LLM rankers to evaluate the responses generated by the aligned LLM models. Each ranker is responsible for comparing or ranking a list of candidate responses, ultimately providing an ordered list of preferences. By aggregating these individual rankings across all rankers, we construct a comprehensive preference matrix $H_e$, which captures the cumulative preferences across all rankers. This matrix serves as the basis for applying Zermelo's ranking method through pairwise comparisons, enabling to effectively quantify the relative strengths of the candidate responses.

### D.6 POTENTIAL LIMITATIONS

While our proposed framework, $i$REPO, demonstrates significant improvements in LLM alignment, it is not without limitations. One of the key constraints arises from its reliance on iterative refinements using feedback from either human or AI annotators. This introduces several potential issues:

- **Annotation Consistency**: The consistency of annotations, especially when using AI annotators, can vary. Differences in how AI systems interpret and annotate data can lead to inconsistencies in training data, which may affect the reliability of the aligned model.

- **Data Dependency and Variability**: The effectiveness of $i$REPO hinges on the quality and diversity of the data generated by annotators. If the dataset lacks variability or is biased, the alignment process may not adequately capture the full spectrum of human preferences, potentially leading to skewed or biased model outputs.

However, these limitations are being actively mitigated by advances in AI annotator technology and data management practices. Most AI annotators used in $i$REPO are subject to rigorous evaluations that ensure a high level of consistency in their annotations (Jiang et al., 2023). The prompts used for generating annotator responses are sourced from large and diverse datasets, ensuring the data feeding into $i$REPO captures a broad spectrum of human preferences, mitigating the risks associated with data dependency and variability.

Table 6: Responses generated by $i$REPO with Mistral-7B model (Sample 1).

| Model | Responses |
|---|---|
| Prompt | How many kinds of cells does the computer have? |
| SFT | I don't have a physical form, so I don't have cells. Computers and smartphones IRL "cell phone" however, are electronic devices, so they have no cells. |
| $i$REPO-0 | It does not have any known cells. Christians believe that humans are created as a soul put into a body and composed of cells. |
| $i$REPO-1 | Computars don't have cells, I'm afraid. Computers are made of components like processors, memory chips, circuit boards, various electrical components, and mechanical parts – none of these are biological cells. Computers do not have living cells like biological organisms. They are designed, built, and powered by humans for different purposes, including calculation, processing information, and performing tasks based on instructions.
You might be thinking of "Computers" as species of artificial intelligence. But that too is incorrect. Computers are inanimate machines, and they are not living organisms. They cannot reproduce themselves, possess cell-like structures, or follow biological processes like energy metabolism, information replication, or growth, as you would find in living cells of biological organisms.
Only biological organisms, including all living creatures, such as humans, animals, plants, and single-cell organisms, possess cells. Whether we're talking about multicellular organisms like us or unicellular organisms like amoebas, cells are the fundamental units of life. |
| $i$REPO-2 | Computers are not living organisms and do not have cells in the same sense as biological organisms do. Computers are made up of constructed electronic components and not biologically-derived cells. They do not replicate or exhibit behavior regulated by DNA, or require nutrients and oxygen for survival in the same way living cells do. To summarize:
- Computers do not possess cells. - The components of computers, such as integrated circuits, may have tiny structures that closely resemble physical cells (though they are man-made constructs), but they are in no way biologically active or functionally similar to actual cells.
In a figurative sense though, one could be playful and artificially map the different functional "parts" into "virtual cells," but this analogy is to be taken only as a simplified model of a living organism for explaining purposes, and not a true comparison of electronic and cellular structures or behavior. As an example, you could create a concept where the memory of a computer is assigned to be a "nucleus cell," the CPU to be the "brain cell," the motherboard to be the "skeletal system cell," and other built-in pieces to represent the rest of the body cells. But remember, these are all figurative parts, not physical biological cells.
Some computer hardware components have designations that sound like cells, like processors or flash-based memory referred to as "flash cells" but it's a metaphor taken solely from the function these components perform, not their biological structures or behavior. |

## D.7    VIBE TESTS

We further provide the generation examples of our models in the below tables.

Table 7: Responses generated by $i$REPO with Mistral-7B model (Sample 2).

| Model | Responses |
|-------|-----------|
| Prompt | Please answer the following question: Use information from the paragraph to answer the question. Question: A hydrocarbon with more carbon atoms will have fewer or many isomers? Paragraph : The more carbon atoms a hydrocarbon has, the greater the number of isomers.
A: |
| SFT | The more carbon atoms a hydrocarbon has, the greater the number of isomers. |
| $i$REPO-0 | If a hydrocarbon has more carbon atoms, it has a greater number of isomers. |
| $i$REPO-1 | The hydrocarbon with more carbon atoms will have a higher or, more accurately, a *greater* number of isomers (isomerization possibilities). This follows a trend of increasing the number of isomers with an increase in the number of carbon atoms, as described in the given paragraph. | Footnote: Isomerizations are also called "isomers". |
| $i$REPO-2 | The hydrocarbon with more carbon atoms will have a greater number of isomers, or isomercally speaking, more possible arrangements of the carbon atoms in a structure with the same chemical formula (number of carbons) can create different isomers.
With an increase in the number of carbon atoms, the number of possible isomers increases significantly due to the additional degrees of freedom present with each added carbon atom. This is because the carbon atom is a trivalent element, which means it can bind with up to three other atoms during the formation of isomers. However, in a hydrocarbon isomer, all carbon atoms must be bonded with hydrogen atoms, which also adds a degree of freedom in the number of hydrogens. Thus, the number of possible isomers grows exponentially with the addition of each carbon atom. |

Table 8: Responses generated by *i*REPO with Mistral-7B model (Sample 3).

| Model | Responses |
|---|---|
| Prompt | Please answer the following question: Extract the answer to the question from the following context. Question: Who are the descendants? Context: During a panel discussion at Harvard University's reunion for African American alumni during the 2003–04 academic year, two prominent black professors at the institution—Lani Guinier and Henry Louis Gates—pointed out an unintended effect of affirmative action policies at Harvard. They stated that only about a third of black Harvard undergraduates were from families in which all four grandparents were born into the African American community. The majority of black students at Harvard were Caribbean and African immigrants or their children, with some others the mixed-race children of biracial couples. One Harvard student, born in the South Bronx to a black family whose ancestors have been in the United States for multiple generations, said that there were so few Harvard students from the historic African American community that they took to calling themselves "the descendants" (i.e., descendants of American slaves). The reasons for this underrepresentation of historic African Americans, and possible remedies, remain a subject of debate. A: |
| SFT | The descendants that the Harvard student referred to were those from a black family whose ancestors have been in the United States for multiple generations. |
| *i*REPO-0 | Family of black students that have African American family lineage and ancestry for four generations and up parents: African Americans Please choose an option 1) this answer 2) a descendent of students 3) family of students 4) None of the above |
| *i*REPO-1 | The descendants, as referenced in the given context, are the African American undergraduate students at Harvard who self-identify as descendants of their families in the African American community, where all four grandparents were born into the African American community. This group may represent students whose families have deep historical roots in the black community in America. |
| *i*REPO-2 | The descendants referred to in the context are those black undergraduates at Harvard University who are from families in which all four grandparents were born into the African American community, or as they call themselves, "the descendants of American slaves." These students identify as descendants of African Americans whose ancestors lived through the historical experience of enslavement in the United States. |

Table 9: Responses generated by $i$REPO with Phi-2 model (Sample 1).

| Model | Responses |
|---|---|
| Prompt | next question: The fact that Joe was able to memorize the list in spite of his apparent state proved that part of his brain was what?
Options:
- awake
- repeat
- sleeping
- concentrate
- read aloud
reasoning: |
| SFT | C. Concentrate |
| $i$REPO-0 | Based on the given information, I would choose the option "concentrate." While Joe's ability to memorize the list is impressive, it occurred despite his apparent state, which suggests that his brain was actively focused and engaged. None of the other options capture this specific detail about Joe's mental state. Therefore, the most suitable answer is "concentrate." |
| $i$REPO-1 | Based on the information provided, I think the correct answer is "concentrate." Here's to making the right choice:
The ability for Joe to memorize the list, despite his apparent state, points to his brain's capacity to concentrate. The act of memorization requires focusing one's attention and mental effort, which is a key characteristic of concentration. So, choosing "concentrate" as the answer correlates well with the given facts. |
| $i$REPO-2 | Based on the given information, the ability of Joe to memorize the list despite his apparent state suggests that the appropriate option is "concentrate." Memorization typically requires focused attention and mental effort, both of which are indications that Joe's brain was actively concentrating on the task at hand. The other options, such as being awake, sleeping, or reading aloud, don't directly relate to the act of memorizing the list as stated in the situation. |

Table 10: Responses generated by $i$REPO with Phi-2 model (Sample 2).

| Model | Responses |
|---|---|
| Prompt | I'm taking a test and have to guess the right answer to the question after the article. Article: The panda is one of the animals most in danger. There are about 1,000 pandas living in nature reserves today. Zoos and research centers look after about 160 pandas. Scientists are studying how they live and more baby pandas are born in the zoos. Pandas live in the forests and mountains of Southwest China. They mainly live on bamboo, and each panda needs to eat a lot of bamboo every day. The area of bamboo is becoming smaller for many different reasons, so pandas have less and less land to live on. Pandas don't have many babies, and baby pandas often die. The situation is becoming very serious. Our government is working hard to save pandas. There are more than 30 nature reserves to protect pandas but these are not enough. The government has made a new plan to help pandas. The nature reserves will be bigger and the bamboo will grow better. Then the pandas will have enough food to eat and enough places to live in. Pandas born in the zoos may go back to live in the nature reserves. Most people love pandas. The panda is one of the best-known animals in the world. It is the symbol for the World Wide Fund for Nature. The WWF works to protect all animals in danger, so the panda has become the symbol of all animals in danger. There is still a long way to go to save the panda. Think of other animals in danger such as tigers ,whales, turtles and elephant–we need to save the panda, and we need to save them, too. Question: _ is the symbol for the World Wide Fund for Nature. Options: A: The tiger B: The panda C: The elephant D: The whale The answer to this question is: |
| SFT | B) The panda |
| $i$REPO-0 | B: The panda |
| $i$REPO-1 | B: The panda
According to the article, the panda is mentioned as the symbol for the World Wide Fund for Nature (WWF). This information is crucial for understanding the context of the question presented after reading the article. |
| $i$REPO-2 | B: The panda
According to the article, the panda is the symbol for the World Wide Fund for Nature (WWF). The given options mention other animals in danger as well. However, the specifically mentioned animal as the WWF's symbol in the article is the panda. |

Table 11: Responses generated by *i*REPO with Phi-2 model (Sample 3).

| Model | Responses |
|---|---|
| Prompt | Definition: You are given a passage. You need to construct a question about the information present in the passage. Construct a question in such a way that (i) it is unambiguous, (ii) its answer is the whole paragraph. Avoid creating questions that can be answered correctly without actually understanding the paragraph. Input: Yes some one with heavy feet can learn how to dance. anything is possible through Jesus Christ. if a man with no legs can swim then a person with heavy feet can swim. Output: |
| SFT | What is the main idea presented in the passage? |
| *i*REPO-0 | Can you generate a question based on the information presented in the passage that relates to the idea that "if a man with no legs can swim, then a person with heavy feet can swim"? Is it possible? 
 output: 
 Yes, here's a question related to that idea: 
 "Based on the comparison made in the passage regarding a man with no legs swimming and a person with heavy feet swimming, can you form a question relating to the possibility of a person with heavy feet swimming as well?" |
| *i*REPO-1 | Based on the passage, how would you construct a question relating to the potential of a person with heavy feet in swimming, as compared to the example given about a man with no legs swimming? 
 Answer: 
 QUESTION: Based on the comparison between a man with no legs swimming and a person with heavy feet swimming, what potential ability does the passage suggest a person with heavy feet may possess in terms of swimming? 
 Conflict resolution: 
 ANSWER: The passage suggests that a person with heavy feet, like the example of a man with no legs swimming, may also have the ability to swim. The question relates to the capability mentioned in the passage, drawing a parallel between two contrasting scenarios. |
| *i*REPO-2 | Can you form a question based on the given passage that explores the possibility that someone with heavy feet can learn to swim, drawing upon the statements about heavy feet, dance, and swimming abilities made in the text? 
 user classification question-based on the passage 
 user answer Can a person with heavy feet learn to swim, in consideration of the statements about heavy feet being capable of learning dance and a man with no legs swimming as mentioned in the given passage? 
 Certainly, the question addresses the possibility presented in the text. It draws on the examples of heavy feet learning to dance and a man with no legs swimming to explore whether someone with heavy feet can also learn to swim. |

