# OpenReview forum: "$i$REPO: $i$mplicit Reward Pairwise Difference based Empirical Preference Optimization"
_ICLR.cc/2025/Conference — ICLR 2025 Conference Withdrawn Submission_

### Official Review · Reviewer_pqS1 · 2024-10-31

**Soundness:** 2
**Presentation:** 3
**Contribution:** 2
**Rating:** 3
**Confidence:** 4

**Summary:**

This paper studies a method to do the policy optimization for language models. The authors leverage an empirical estimation of the win rate $w_{ij}$ and use the square loss to optimize the policy function. The authors showed both theoretical analysis and empirical algorithm with validation.

**Strengths:**

- This paper is well written and easy to follow with
- The authors provide some of the theoretical justification for the performance of the algorithms
- Empirical validation shows the performance of the algorithm

**Weaknesses:**

My major concern is the contribution of the algorithm and the theoretical analysis listed as below

- In section 4.1, the delivered message, specifically $w_i / (w_i + w_j) = h_{ij} / (h_{ij} + h_{ji})$ is somewhat trivial. The rationale behind highlighting this equivalence is unclear, as it does not seem to directly address any core technical challenges. A stronger connection to the problem statement or contribution would help clarify the authors’ intent here.
- In section 4.2, The proposed method employs the square loss to approximate the binary distribution, defined as $L = E[(p - p^)^2]$. The authors do not provide a clear reason for selecting this loss function over the more conventional KL divergence, $L = E(\mathrm{KL}(p \parallel p^))$, which is typically a “soft” version of cross-entropy loss. It would be beneficial for the authors to clarify why KL divergence was deemed unsuitable for this application and how the square loss better serves the purpose of approximating the distribution $p$.
- In addition, the algorithm requires repeated evaluations of $h$ for each individual preference pair $(i, j)$. The authors should clarify why this $h$-step data cannot be directly used within DPO but instead must be aggregated into a squared loss function. For fair comparison, given that $h = 5, 9, 15, 21$ were chosen in the appendix, the iterative DPO dataset should ideally leverage these same values ($h = 5, 9, 15, 21$) rather than a single annotator in vanilla iterative DPO. A clarification on this point is necessary.
- In section 4.3, the theoretical justification here requires further detail. For example, Assumption 4.1 cannot hold universally for any $P_h$. The authors may want to define the releasability on the ground truth $P^$, indicating that $P_h \to P^$ as $h \to \infty$ (per Assumption 4.2). However, this assumption may be also impractical, as it requires a significantly large $h$. Moreover, the theoretical analysis as presented is insufficient. For example, equation (18) essentially states that if $L = (p - q)^2 = 0$, then $D_{TV}(p, q) = 0$, which is self-evident. Similarly, equation (19) mirrors the DPO analysis. Furthermore, the role of $C_t$ in Theorem 4.5 is not well-explained, especially within the online iterative framework. A more thorough explanation of these theoretical points would strengthen the analysis.
- The authors should explicit show the selection of $h$ in the main paper.

### Regarding the proof in the appendix, here're a few issues I'm suggesting to revise.
- When applying the mean-value theorem, the correct expression is $|\sigma(x_1) - \sigma(x_2)| = \Big|\sigma’(x_0)\Big| \times  |x_1 - x_2|$ (with absolute value of $|\sigma'(x_0)|$ rather than simply $\sigma’(x_0)|x_1 - x_2|$, though this does not affect the correctness of the proof.
- In line 810, this is a concentration inequality rather than the law of large numbers. The scaling factor of $O(1 / \sqrt{n})$ relies on certain conditions, which should be explicitly verified by the authors (e.g. Hoeffding's inequality with R-sub-Gaussian). The law of large numbers only guarantees convergence to zero but does not provide scale guarantees.

**Questions:**

Please see weakness

---

### Official Review · Reviewer_tWpc · 2024-11-01

**Soundness:** 4
**Presentation:** 3
**Contribution:** 2
**Rating:** 3
**Confidence:** 4

**Summary:**

The paper proposes a regression based loss function for LLM alignment. This loss function leverages the probability of human annotators ranking one response higher than another to construct soft lables. Through empirical experiments, the authors demonstrate that the regression based loss function leads to better alignment in comparison to DPO and IPO. The authors use Phi-2 and Mistral-7B as the base model. They perform experiments using the LLM-eval-harness and Multi-turn evaluation frameworks. The authors also theoretically demonstrate that their method achieves optimal results under certain contributions

**Strengths:**

1. The empirical results show that iRepo outperforms both DPO and IPO.
2. The ablation studies show that leveraging logits does lead to better alignment. Further ablation studies also show the effect of the number of annotators used to generate the logits.
3. The theoretical analysis is clear even though the assumptions and conditions are likely too specific.

**Weaknesses:**

1. The authors do not address the additional computational costs to update the weights $w_i$ in each iteration. The comparison between DPO/IPO and iRepo should take into account this increase in time complexity
2. A fundamental limitation of this method is that it requires multiple annotators for each pair of output. This is not applicable to DPO/IPO.
3. There seems to be a disconnect between the description of DPO and IPO across multiple sections. Until section 3, the authors refer to vanilla DPO/IPO. In the experiments, the authors use iterative DPO/IPO as the baseline without describing the iterative version of the algorithm.
4. It seems like simply increasing the number of annotators does not lead to better alignment. Infact, this seems like a strange hyperparameter to tune. Given the marginal improvement over the baselines, Figure 2(b) needs further explanation/investigation.
5. I would encourage the authors to add stronger baselines apart from DPO/IPO: SPPO, KTO, SimPO etc.

**Questions:**

1. Have you considered using the raw probability generated using annotators: $P(y_s > y_l | x) = 1/h * \sum_{H} I[ y_s > y_l ] $

---

### Official Review · Reviewer_tx8F · 2024-11-03

**Soundness:** 2
**Presentation:** 3
**Contribution:** 2
**Rating:** 5
**Confidence:** 4

**Summary:**

This paper proposes iREPO, which leverages the Zermelo’s model to construct empirical human preference. With this scalar preference target, iREPO regresses the implicit reward pairwise difference towards the logit of empirical human preference, allowing effective LLM alignment. The proposed method is supported with theoretical analysis and empirical results.

**Strengths:**

* The presentation of the paper is clear and easy to follow.
* Empirical results validate the proposed algorithm.
* Theoretical analysis is provided for the proposed method.

**Weaknesses:**

* Correctness.
  * Line 042: online and offline alignment. Online alignment does not necessarily involve RLHF based on PPO. In fact, typical RLHF approaches are instances of offline RL, where a reward model is learned using offline dataset then an RL algorithm learns the policy (LLM) using that learned reward.
* Motivation of Zermelo's Model. Zermelo's algorthm is used to estimate the target regression values of implicit reward (eq16). In my opinion, it's also possible to leverage any reward model (trained with eq2) to provide such regression target. Could you elaborate why Zermelo's model is proposed and what is its advantages?
* Minor issues. Grammar or spelling errors: line 011 (Large), 108 (LLM Alignment), 253, 852.

**Questions:**

1. Can you elaborate "Conventional approaches often approximate this reward model through reinforcement learning techniques (Williams, 1992; Ahmadian et al., 2024; Schulman et al., 2017)." at line 200? I don't get how these references apply RL to approximate the reward model.
2. In Table 3, iREPO-2 is compared to Iter-DPO & Iter-IPO. Which version of Iter-DPO/IPO is used? Could you give a full results similar to Table 2, showing different stages of Iter-DPO/IPO?
3. Which specific nine LLM rankers did you adopt to simulate human preferences?

---

### Official Review · Reviewer_8Srs · 2024-11-04

**Soundness:** 2
**Presentation:** 3
**Contribution:** 2
**Rating:** 3
**Confidence:** 4

**Summary:**

The paper contributes a new method for aligning language models with human/AI feedback. Assuming multiple responses can be compared, the model proposes to compute a quantity indicating how strongly humans prefer one answer over the other, and use a DPO/IPO like objective regress onto the strength. The paper provides a principled approach for computing the strength of preference, and shows some empirical evidence showing the utility of this.

**Strengths:**

The paper makes an interesting contribution for (a) computing the strength of preference when given two completions and (b) providing a principled framework to compute such relative preference strength. The writing is reasonably clear, and the paper provides substantial theory to go along with their algorithm.

**Weaknesses:**

- Currently, the annotation scheme is rather expensive. Collecting a single preference pair requires sampling D completions, making D choose 2 pair and annotating each pair multiple times. After computing the logit corresponding to the strength of the preference, all other completions are thrown away.
- The experimental setup can be improved and made fairer (questions below). First, the main contribution of the paper is a new offline objective for alignment. A more informative experiment would be to compare the method first in a completely offline setting as follows: Take a SFT model, sample D completions per prompt, get preference labels for each pair. Use iREPO to regress onto the strength of preference (with and without filtering down to weakest/strongest pair), but there are many ways to compute ranking here and you could consider putting in all the preferences into DPO/IPO too (as they are implicitly ranking losses).
- More experimental evidence would help, as a lot of these evaluations are fairly noisy.
- In general, the paper needs to justify why expending this additional effort on collecting preference label is justified. The claim can be backed by showing better performance / unit of annotation effort. Or you could show that these preference labels result in higher signal-to-noise ratio for preferences, and can be scaled better.

**Questions:**

- Do you need always need a dense $$H_e$$ matrix. This would imply that you need to observe at least D choose 2 comparison * number of annotators? Since, iREPO filters down comparison to strongest vs weakest comparison, could you reduce the annotation burden?
- Since you are computing the logit to regress to, why not regress to the logit for all pairs of comparisons instead of filtering down to the comparison of the strongest and weakest completion pair? For other methods where strength of preference cannot be determined, filtering makes sense as they are treated similarly (with a hard preference label). However, since you have notion of "strength" of preference that you are regressing to, I'd be curious to see the results where you do not filter preference pairs.
- UltraFeedback and other datasets are based on GPT-4 labels, does labeling with DeBERTa introduces a mismatch in the preference function, especially with MT-Bench with uses GPT-4 to evaluate model preferences again?
- Could you elaborate how the new labels collected, during iteration 1 and 2? How are the labels computed for iter-DPO and iter-IPO? I am worried the comparison isn't fair.
- Could you provide more detail on how the number of annotators change the preference labels/performance of iREPO?

---

### Official Review · Reviewer_pogN · 2024-11-04

**Soundness:** 2
**Presentation:** 3
**Contribution:** 1
**Rating:** 3
**Confidence:** 3

**Summary:**

The paper presents iREPO, an empirical preference optimization framework for language model alignment, grounded in theoretical exploration with an emphasis on iterative feedback mechanisms. While the paper is structured well and includes clear algorithmic steps, the experimental design is weak, as iREPO, an online method, is primarily compared to offline baselines. This choice undermines the validity of the results, as offline methods generally face limitations in out-of-distribution scenarios, which are not an issue for online methods. Furthermore, the use of non-state-of-the-art models like Phi-2 and Mistral-7B weakens the contribution, making it difficult to assess iREPO’s potential on current high-performing models such as LLaMA 3. The approach lacks novelty, offering incremental rather than transformative improvements to preference optimization. Key questions remain, particularly regarding the rationale for comparing iREPO with offline methods, how it would perform on state-of-the-art models, and its robustness in real-world settings with distributional shifts. Overall, the paper falls short in demonstrating strong empirical evidence and addressing practical concerns for broader applicability.

**Strengths:**

1. Theoretical Exploration: The theoretical exploration around implicit reward pairwise differences is well-motivated, and the authors provide proofs and assumptions to support their claims.
2. Clarity in Model Steps: The procedural flow in the algorithm (iREPO) is clearly outlined, making the technical contributions easy to follow for researchers familiar with preference optimization.

**Weaknesses:**

1. Comparative Analysis: The choice to compare an online method to several offline methods (e.g., DPO, IPO) is questionable and potentially misleading, as offline methods generally face limitations in out-of-distribution scenarios that online approaches can better handle. This mismatch in methodology reduces the validity of the experimental claims.
2. Model Selection: The experiments are conducted only on Phi-2 and Mistral-7B, which are not state-of-the-art models. To demonstrate iREPO’s robustness and generalizability, it would be more insightful to test on current high-performing models like LLaMA 3. Without this, the results are less relevant to ongoing research in language model alignment.
3. Limited Real-World Applicability: Given the reliance on certain assumptions (e.g., no data distribution disparity), the applicability of iREPO in real-world, dynamic settings remains uncertain.

**Questions:**

1. How would iREPO perform on state-of-the-art models? Since only outdated models were used, testing iREPO on LLaMA 3 or similar state-of-the-art models would be crucial for determining its broader impact.
2. Why were offline methods chosen as primary baselines? A more robust comparison with other online methods would provide a fairer evaluation of iREPO’s capabilities. The authors should address the rationale for focusing on offline baselines.
3. Does iREPO maintain effectiveness under distribution shifts? Given that distributional shifts are common in real-world scenarios, an exploration of iREPO’s resilience to these changes would enhance the study's practical relevance.

---

### Note · Authors · 2024-11-26

I have read and agree with the venue's withdrawal policy on behalf of myself and my co-authors.